# CCRRSleepNet: A Hybrid Relational Inductive Biases Network for Automatic Sleep Stage Classification on Raw Single-Channel EEG

**DOI:** 10.3390/brainsci11040456

**Published:** 2021-04-02

**Authors:** Wenpeng Neng, Jun Lu, Lei Xu

**Affiliations:** 1College of Computer Science and Technology, Heilongjiang University, Harbin 150080, China; 2181397@s.hlju.edu.cn (W.N.); 2181387@s.hlju.edu.cn (L.X.); 2Key Laboratory of Database and Parallel Computing of Heilongjiang Province, Heilongjiang University, Harbin 150080, China

**Keywords:** deep learning, automatic sleep stage classification, inductive biases, relational inductive biases

## Abstract

In the inference process of existing deep learning models, it is usually necessary to process the input data level-wise, and impose a corresponding relational inductive bias on each level. This kind of relational inductive bias determines the theoretical performance upper limit of the deep learning method. In the field of sleep stage classification, only a single relational inductive bias is adopted at the same level in the mainstream methods based on deep learning. This will make the feature extraction method of deep learning incomplete and limit the performance of the method. In view of the above problems, a novel deep learning model based on hybrid relational inductive biases is proposed in this paper. It is called CCRRSleepNet. The model divides the single channel Electroencephalogram (EEG) data into three levels: frame, epoch, and sequence. It applies hybrid relational inductive biases from many aspects based on three levels. Meanwhile, multiscale atrous convolution block (MSACB) is adopted in CCRRSleepNet to learn the features of different attributes. However, in practice, the actual performance of the deep learning model depends on the nonrelational inductive biases, so a variety of matching nonrelational inductive biases are adopted in this paper to optimize CCRRSleepNet. The CCRRSleepNet is tested on the Fpz-Cz and Pz-Oz channel data of the Sleep-EDF dataset. The experimental results show that the method proposed in this paper is superior to many existing methods.

## 1. Introduction

In recent years, the performance of automatic sleep stage classification algorithms based on deep learning has gradually surpassed the performance of traditional machine learning methods and human experts. Deep learning methods often follow the end-to-end design concept, and emphasize the minimum prior representation and computational assumptions [1]. This minimum prior representation and computational assumption can be represented by inductive biases. Whether the inductive biases of the algorithm match the problem itself in most cases directly determines whether the algorithm can achieve good performance. Inductive biases can be divided into relational inductive biases and nonrelational inductive biases. The relational inductive biases are the key to determining the upper limit of the theoretical performance of the algorithm, and the nonrelational inductive biases determine the extent to which the actual performance reaches the theoretical performance. In deep learning, the connection of neurons reflects the relational inductive biases, such as convolution layer of local connection and recurrent layer of cross time step connection. The nonrelational inductive bias is reflected in other aspects, such as activation function, standardization, data augmentation, optimization algorithms, etc.

Currently, deep learning-based algorithms usually use epoch-based models and sequence-based models. The epoch represents a 20- or 30-s period of polysomnography (PSG) data with a label. It is the smallest segment divided by human experts on sleep signals. The epoch-based model assumes that the sleep phases are independent and identically distributed, and uses a structure with certain relational inductive biases for representation learning. One approach is that the epoch level signal has only locality and translation invariance and uses convolutional neural network (CNN) for representation learning. For example, Manzano et al. [2] used one-dimensional CNN to achieve an overall accuracy of 68.9% on the University College Dublin Sleep Apnea Database (UCD) dataset. Arnaud et al. [3] used one-dimensional CNN to achieve an overall accuracy of 87% for the Sleep Heart Health Study (SHHS) dataset. Orestis et al. [4] adopted one-dimensional CNN and then further used two-dimensional CNN to extract associated features. Adding additional relational induction biases based on locality and translation invariance can extract simpler and more efficient features, and make the network easier to optimize. An overall accuracy of 74% of the Fpz-Cz channel of the Sleep-EDF dataset is achieved. Zhang et al. [5] extended CNN to the complex domain, imposed the constraint of the decision boundary of the real and imaginary parts orthogonal to the convolution kernel, and achieved an overall accuracy of 92% on the Harvard-MIT Division of Health Sciences and Technology (MIT-BIH) dataset. Zhang et al. [6] used orthogonal convolution to impose orthogonal constraints on the convolution kernel and achieved an overall accuracy of 88.4%on the UCD dataset and 87.6% on the MIT-BIH dataset. Ahmed et al. [7] and Alexander et al. [8] used jump connections to impose constraints on the topology and improve the information flow and gradient flow of the network. Huy et al. [9] adopted the pooling strategy of 1-Max Pooling CNN to improve the shortcomings of ordinary pooling for information loss and achieved an overall accuracy of 82.6% on the Sleep-EDF dataset. Although these methods captured the differences between sleep stages to varying degrees, the hypothesis of independent and identical distribution was difficult to establish, so this method had natural defects.

The sequence-based model believes that physiological signals not only show different characteristics in sleep stages, but also have state transition relationships between sleep stages. Therefore, after the epoch-based model processing, the transformation relationship learning algorithm between sleep stages is added to learn the salient features in sleep stage and the implicit transformation relationship between sleep stages. Stanislas et al. performed feature fusion of adjacent K epochs [10]. Cui et al. optimized the sequence length based on fine granularity and achieved an overall accuracy of 92.2% for the ISRUC-Sleep dataset [11]. Zhang et al. extracted epoch features from wearable device data, and used Bi-LSTM to learn sequence relations [12]. Supretak et al. proposed DeepSleepNet, which used CNN for epoch representation learning in the first stage and Bi-LSTM for sequence learning in the second stage [13]. It achieved an overall accuracy of 82% in the Fpz-Cz channel of the Sleep-EDF dataset and the overall accuracy rate was 79.8% in the PZ-Oz channel. Huy et al. proposed a SeqSleepNet with two-layer Bi-GRU for sequence-to-sequence learning and achieved an overall accuracy of 87.1% for the Montreal Archive of Sleep Studies (MASS) dataset [14].

Although the current epoch-based model and sequence-based model have made some progress, the epoch-based model needs to have a strong assumption of independent and identical distribution between epochs, which makes it impossible to obtain the potential connections between epochs. Due to the insufficient detail in the division of signal levels in the model, the information contained in the signals at the same level is very complicated. It is difficult to completely extract the information using only a single relational inductive bias. For example, the signal divided into fragments contains time-varying characteristics, and the simple translation invariance relationship induction bias cannot obtain all of its information. Therefore, a hybrid relational inductive biases network is proposed in this paper. It is called CCRRSleepNet. Specifically, in the sequence-based model, this paper further subdivides the physiological signal into three levels in the time domain, frame, epoch, and sequence, as shown in Figure 1.

In Figure 1, the signal divided by the red vertical line is an epoch; 30-s PSG data are adopted in this paper. An epoch can continue to be divided into frames (green vertical line segmentation), and all epochs constitute a sequence. The EEG signal is a nonstationary and nonlinear random signal. A simple processing strategy is to treat the EEG signal as composed of very small stationary periodic signals (i.e., frames). This paper divides the signal into frames and uses the translational invariance relational inductive biases to learn time-invariant features. The time-invariant relational inductive biases are used to learn the time-varying features between frames. A complete sequence contains multiple 30-s epochs, which requires the use of time-invariant relationship induction bias to learn the transformation relationship between different epochs, that is, the latent change law between sleep states.

More detailed levels are conducive to applying matching and simple relational inductive biases on each level, and the corresponding features are easy to extract. However, there are still two problems: first, the time domain hierarchical division does not carry out feature decomposition, so a single structure cannot obtain all the useful information. Second, even for the same feature type, there is a parameter selection problem when the corresponding structure is used. The performance difference caused by different parameters is very large. For example, the size of the convolution kernel of CNN directly affects the acquisition of signal features. Therefore, a variety of relational inductive biases hybrid methods at the same level are adopted in this paper, and time-varying and time-invariant features in the signal are extracted to the maximum extent. Multiscale atrous convolution block (MSACB) was used to enhance the capability of CNN to extract complex signal features.

The nonrelational inductive biases determine whether the algorithm can reach the theoretical optimal value. Deep learning algorithms are data-driven. If the true distribution of the training data and the test data is quite different, it will cause serious overfitting problems. There are some problems in terms of sleep signals: (1) The number of samples among different classes is extremely unbalanced; (2) there is a significant difference in the difficulty of distinguishing among similar samples; (3) the number of transformation relationships between state transitions is unbalanced. In order to solve the above problems, a method based on sequence sampling is proposed to balance the state transition relationship. The focal loss [15] function is used to solve class imbalances and mine difficult samples. In addition, the Mish activation function [16] is more tolerant of the negative values of the signal. The AMSGrad optimizer is adopted to distinguish the contributions of different batches.

In summary, the contributions of this paper include the following:

An end-to-end hybrid relational inductive biases network, CCRRSleepNet, is proposed, which is used for automatic sleep stage classification. It can alleviate the algorithm performance ceiling caused by insufficient feature extraction methods.A multiscale atrous convolution block (MSACB) is proposed, which uses fewer parameters and enhances the ability to extract complex signal features.A variety of nonrelational inductive biases are used to optimize the network. A sequence-based sampling method is used to balance the transition between sleep states. The focal loss function is used to alleviate the challenges of class imbalances and difficult samples. A nonmonotonic, overall smooth Mish activation function that allows a small number of negative values to pass through is used. It can stabilize the training process. Experiments show that these methods can improve the performance or network training process.The proposed model uses a single channel EEG as input, and EEG does not require any processing. Experiments show that the proposed method has significant effects on Fpz-Cz and Pz-Oz channels of Sleep-EDF datasets.

## 2. Methods

### 2.1. Relational Inductive Biases

To solve the problem of incomplete feature extraction in automatic sleep stage classification tasks, a hybrid relational inductive biases network called CCRRSleepNet is proposed, as shown in Figure 2. Our open-source code can be found here: https://github.com/nengwp/CCRRSleepNet (accessed on 2 April 2021).

CCRRSleepNet is an end-to-end network, which is composed of three parts: (a) Frame-level convolutional neural network: this part mainly extracts some low-level time-invariant features of the signal, such as amplitude, skewness, slope, phase, and other basic attributes; (b) Epoch-level hybrid neural network: a mixture of a convolutional neural network and a recurrent neural network. It is used to extract advanced time-invariant features and advanced time-varying features in signals. (c) Sequence-level recurrent neural network: uses the recurrent neural network to obtain the transformation relationship between sleep stages. It optimizes the sleep stages based on the sequence relationship. In summary, the network is a hybrid of frame-level CNN, epoch-level CNN/Recurrent neural network (RNN), and sequence-level RNN. The network structure can be abbreviated as C-C/R-R. Further abbreviated and related to the task. The network can be called CCRRSleepNet. The input of the model is an unprocessed sleep signal. Although the experiment in this paper uses a single-channel signal, the model in this paper supports multichannel signals and supports different sampling rates.

#### 2.1.1. Frame-Level Convolutional Neural Network

The nonstationary time-varying EEG signal can be regarded as composed of many very small periodic stationary subsignals, so the time-varying characteristics of the signal in one frame can be ignored. Firstly, the frame-level CNN uses two very different convolution kernels to obtain signal characteristics at different frequencies. The size of the convolution kernel is related to the sampling rate of the signal and the effective frequency of the EEG; 25 and 100 are adopted in this paper. For convolution kernels of different sizes, a positive correlation step size is used to reduce unnecessary computation, and then the max-pool is performed. In the two branches, the product of the pool kernel size and the step size of the previous convolution are equal. After concatenating the features of CNN from different branches, a dropout with a keep rate of 0.5 is used to prevent overfitting. Secondly, multiscale atrous convolution block is used to extract features. MSACB uses convolution layers with different receptive field sizes and depths to fully extract various frequency components and features of different difficulty in sleep signals. As shown in Figure 3, MSACB has four branches, the number of layers of the branches increases sequentially, and the receptive field is expanded by using atrous convolution in the process of increasing. The convolution layer with a convolution kernel size of 1 adds a layer of nonlinear ability, and uses very few parameters and little computation. Atrous convolution can achieve a larger receptive field with fewer convolution parameters and less computation.

It should be noted that the problem of frame-level segmentation is a very important issue. First, a simple strategy is to continue to segment the epoch with 30 s, and then perform feature extraction for each segmented subepoch (i.e., frame), but this means that the correlation of the signal is simultaneously cut. To alleviate this problem, the window function method with overlap ratio in frequency domain analysis methods such as wavelet transform can be adopted. However, it requires manual repeated confirmation to achieve a more appropriate overlap rate. This is inconsistent with the original intention of deep learning. In this paper, a convolution layer with step size to replace it is used. As the convolution is local, which reflects the idea of frame division, the step size represents the overlap rate to some extent. Therefore, although the frame-level CNN extracts features for the frame level, the results are no different from traditional convolution in terms of the form of expression.

After the frame-level CNN processing, the features are extracted at the frame level. These features will be input into the epoch level, and used to learn different advanced features, such as time-invariant signal and time-varying signal. The frame-level features are transformed by a CNN with convolution kernel size of 1 and the global average pooling (GAP) is sent to the sequence level. This means that the frame level information also plays an important role in the final classification results. In addition, this residual connection also enables the gradient of the backpropagation to directly supervise the training of the frame-level CNN.

#### 2.1.2. Epoch-Level Hybrid Neural Network

The feature of signal at epoch level is very complex, so it is necessary to extract it with a hybrid neural network. For time-invariant features, a very simple single-layer CNN is used for feature extraction. For time-varying features, single-layer Bi-GRU is used for feature extraction. Specifically, the features from the frame level are divided into CNN branches and RNN branches learning time-invariant features and time-varying features, respectively. For each branch, firstly, a convolution with 128 channels and convolution kernel size of 1 is performed. The purpose of this method is to transform the frame-level features into epoch-level features, and a different transformation is needed for the CNN branch and RNN branch. Secondly, the max-pool is performed for CNN branches, while the avg-pool is used for RNN. The reason is that the max-pool is nonlinear, while the avg-pool is smooth. This means that the CNN tends to find the most favorable features for the results, while RNN stores more details for each feature, which is very important for time-varying features. Then, the CNN branch is convolved by the convolution kernel, with the size of the convolution kernel is set to 3 and the number of channels is set to 1024. Finally, the global average pooling (GAP) is used to prevent overfitting and each channel is given more explicit feature meaning. For the RNN branch, Bi-GRU with 512 hidden units is adopted, and the last time step of Bi-GRU is used as output.

#### 2.1.3. Sequence-Level Recurrent Neural Network

The 30-s sleep signals divided at the sequence level contain almost all possible time-invariant properties, so the time-invariant features in the signal are no longer repeated learning at the sequence level. There is a close correlation between sleep stages, which is key to exploring the classification of sleep stages. The 1024 features from the frame level and the 2048 features from the epoch level (including 1024 features from the epoch-level CNN and 1024 features from the epoch-level RNN) are concatenated, and a two-layer Bi LSTM with 512 hidden units is used to capture this potential relationship. That is to say, the forward LSTM and the backward LSTM are used to obtain information from the past and the future, respectively. The input from the epoch level is transformed by a full connection with 1024 hidden units, and then added point by point with the output of LSTM to form the residual connection. The final features are classified through a single-layer full connection, and the classification probability value is output through the softmax function. To prevent overfitting, a dropout policy with a keep rate of 0.5 is added to each stage.

### 2.2. Nonrelational Inductive Biases

Nonrelational inductive biases determine whether the learning algorithm can achieve the theoretical best performance. The learning algorithm expects unbiased data for learning, but there is a serious imbalance in sleep data. This imbalance can be divided into three types: (1) The sample number of sleep stage is imbalance; (2) the learning difficulty of samples in the sleep stage is inconsistent, including samples between classes and samples within classes; and (3) the number of state transition relationships between sleep stages is unbalanced. For class sample imbalance, downsampling, upsampling, and generative methods are often used, but these methods bring risks of underfitting, overfitting, and additional computational overhead, respectively. The inconsistency of sample learning difficulty in the sleep stage is reflected in two aspects: (a) Among the samples in the difference classes. For example, the N1 stage is often misclassified. The reason is that the N1 stage is a transitional stage, which has some features in common with the stages before and after the transition. In addition, the N1 stage occurs during a short period and the number is small, so the N1 stage is a difficult sample. (b) The difficulty between samples within a class is different, mainly affected by the significance of features within the class and noise. In this paper, focal loss function is used to solve the problem of class imbalance and inconsistent learning difficulty at the same time. Specifically, focal loss is defined as in Equation (1):(1)$$FL(p)=-\alpha y{\left(1-p\right)}^{\gamma }\log(p).$$

In Equation (1), *p* is the prediction probability vector, *y* is the real label vector, *α* controls the imbalance between classes, and *γ* controls the difficulty of learning. First, *α* is the weight coefficient between classes, usually the inverse of the occurrence frequency of the class. *γ* represents the learning difficulty factor. When a simple sample can be correctly classified, the prediction probability is greater than 0.5. Although the loss of a single simple sample is small, there are a large number of simple samples. The accumulated loss dominates the entire gradient descent process. The *γ* factor can make the loss of simple samples very small, so that a small number of difficult samples with large losses can dominate the optimization process of the entire network.

Another important issue is the imbalance in the number of state transition relationships between sleep stages. The duration of sleep state is usually greater than the 30-s data divided by experts, at least 1‒2 min, and at most several hours, such as W stage. Dividing sleep data into 30-s stages will result in an enhanced transformation relationship, especially when it contains a large number of W stages. Therefore, balancing the number of state transformation relationships is an important means to maintain the true distribution of data and stabilize the training process. A simple method based on sequence sampling is adopted in this paper. First, it performs random downsampling according to the sequence label, so that all classes have the same value, which is equal to the number of the least class multiplied by the sampling factor. The sampling factor is used to control the degree of overlap when the signal is upsampled. Secondly, neighborhood upsampling is performed on these signals: that is, for each signal, multiple epochs in the neighborhood of this signal are randomly intercepted, where the neighborhood length is generally the sequence-level length. After such sampling, the conversion relationship between states can be balanced. Note that the number of classes and the number of state conversion relationships are not absolutely balanced, but this is more in line with the real distribution. Blindly treating the transformation relationship equally will introduce new imbalances.

Inappropriate activation functions may cause problems such as an exploding gradient or vanishing gradient. The signal is more sensitive to negative values, but the current mainstream Relu activation function directly crops negative values, resulting in a certain amount of information loss. Therefore, the Mish activation function is adopted in this paper, which is defined as in Equation (2):(2)$$f(x)=x\times \tanh(\log(1+e^{x})).$$

The activation functions of Mish and Relu are compared in Figure 4. The Mish function has the advantages of being nonmonotonic, passing negative values, and being smooth. 

The experiment in this paper adopts a two-stage training method. First, the frame-level and epoch-level networks are pretrained, and then this part is learned with a smaller learning rate, while the sequence-level part is learned with a larger learning rate. The reason is that the epoch signals need to be trained on the assumption of independent and identical distribution. If the strategy of simultaneous training is used, this assumption will not be satisfied. If there is a problem with the optimization method of the learning algorithm, it may not reach a global minimum but converge to a local minimum. Therefore, it is necessary to decay the weight of the learning rate, and the AMSGrad optimizer is adopted with different contributions for each batch.

Overfitting is an important challenge for learning algorithms. A variety of methods are adopted to alleviate overfitting in this paper. (1) Data augmentation in the pretraining stage. Specifically, random horizontal cyclic shift and random mirror inversion are performed on the signal. This is an important method to alleviate the overfitting of the CNN, but it is not used in the second stage of training. (2) Dropout to prevent overfitting. This means that the model needs to learn more robust features. (3) Global average pooling is adopted to reduce the dimensionality of features in the last layer of the CNN. Traditional fully connected methods are not adopted to avoid overfitting, and to make each convolutional channel learn more univocal features. (4) L2 regularization is applied on the first layer of convolution at the frame level to prevent oversensitivity to sample changes due to overlarge learned parameter values.

## 3. Dataset and Experimental Settings

### 3.1. Dataset

Polysomnography (PSG) is a multiparameter measuring instrument that can record a variety of physiological signals at the same time, such as electroencephalogram (EEG), electrocardiogram (ECG), electrooculogram (EOG), electromyogram (EMG), etc. The PSG data are collected by placing many electrodes and sensors on the patient’s body in the sleep laboratory; these have multiple advantages such as accurate, comprehensive, low-noise, etc. According to the American Academy of Sleep Medicine (AASM) rules [17], PSG data can be divided into five stages: wakefulness (W), rapid eye movement (REM), non-REM 1 (N1), non-REM (N2), and non-REM 3 (N3).

The Sleep-EDF database [18] is often used for benchmark testing of automatic sleep stage classification algorithms. A sleep-cassette subset was published in 2013 by the Sleep-EDF database, containing 39 records from 20 subject. Overnight polysomnographic sleep records from healthy Caucasians aged 25–101 years who did not take sleep-related medications are collected in this subset. Sleep phases in this database were initially labeled by experts according to the R&K standard [19]. S3 and S4 are combined into N3 to comply with the AASM rules in this paper. The EEG data in this database contain Fpz-Cz and Pz-Oz channels with sampling rates of 100 Hz. In this paper, channel Fpz-Cz and channel Pz-Oz are used, and no data preprocessing is involved. The data are divided according to the subjects to ensure that the data of the same subjects do not appear in the training set and the test set at the same time. In order to make a fair and objective comparison with previous studies, W stage data from 30 min before and after sleep are selected. The sample distribution of the Sleep-EDF dataset is listed in Table 1.

As shown in Table 1, the number of different sleep stage samples in the sleep EDF dataset is unbalanced. The neural network updates the weight by optimizing the objective function. In the process of optimization, the contribution of the samples in the high-quantity class and the small quantity class to the optimization objective is the same. However, because the number of samples in the high-quantity class is far more than that in the small-quantity class, the final classification boundary is more inclined to the high-quantity class. This offsets the classification boundary and the final classification performance of the network declines. In this paper, the focal loss function is adopted to adjust the contribution of samples to the optimization objective to alleviate the imbalance problem.

### 3.2. Experimental Settings

The k-fold cross-validation scheme was adopted for experiments. Firstly, the data belonging to the same individual in the sleep EDF dataset is divided into a whole, and then the data of all individuals are divided into κ subsets. In other words, a subset may contain multiple individual data, and the same individual data will only exist in one subset, so that the data of the same individual will not appear in both the training set and the test set at a certain compromise. The κ-th subset was used for testing, and the rest were used as training data. The experiment was repeated k times. Finally, all test results were combined to calculate the final performance. In order to make a fair comparison with other studies, κ was set to 20. The model proposed in this paper used a two-stage training method. First pretraining was performed on the frame-level and epoch-level subnetworks, and then fine-tuning was performed on the entire network. The focal loss function and AMSGrad optimizer were adopted in both stages, where λ, beta1, and beta2 were set to 2, 0.9, and 0.99, respectively. The learning rate decay strategy was used in both stages. In the pretraining stage, the initial learning rate was set to 10^−3^. The learning rate was reduced to 10^−4^ after 40 iterations in the training set, and the pretraining was completed after the next 40 iterations. A fixed learning rate of 10^−6^ was set on the subnetwork part of the fine-tuning stage. The initial learning rate was set to 10^−3^ on the sequence-level part, and then attenuation was performed to 10^−4^, 5 × 10^−5^, and 10^−5^, respectively, every 10 rounds. The batch size was set to 256 in the pretraining stage and 10 in the fine-tuning stage. In order to compare with the DeepSleepNet method, we set the sequence length to 25 epochs. The experimental environment was the TensorFlow deep learning framework based on the Ubuntu operating system. Nvidia GeForce GTX 2080Ti GPU was used for accelerated training. It only takes about 40 min to complete the 1-fold training. Compared with other methods such as DeepSleepNet, SeqSleepNet, and U-time [20] that require several hours of training with the same configuration, the method in this paper has a huge advantage.

## 4. Experimental Results and Analysis

### 4.1. Experimental Results

Fpz-Cz channel and Pz-Oz channel data were adopted to conduct experiments r. In order to comprehensively evaluate the CCRRSleepNet model, a variety of different evaluation indicators were adopted. The evaluation indicators on the per-class level included precision (Pre), recall (Re), and F1-score (F1). The evaluation indicators on the overall level included macro-averaging F1-score (MF1), overall accuracy (ACC), and Cohen’s Kappa coefficient (κ). The experimental results will be introduced and analyzed according to these evaluation indicators.

Table 2 shows the experimental results based on the Fpz-Cz channel data. The table includes a confusion matrix obtained through 20-fold cross-validation, as well as precision (Pre), recall (Re), and F1-score (F1) values for each class. The row label corresponds to the real class, and the column corresponds to the predicted class in the confusion matrix.

As shown in Table 2, the value on the diagonal position of the confusion matrix is much higher than the other values in the row and column of the value. This shows that the CCRRSleepNet model in this paper is effective. From the value of the evaluation indicators, it can be seed that the F1 scores of W, N2, and N3 stages are all above 87, and the F1 score of 82.86 is obtained in the REM stage. At the same time, it should be noted that the N2 stage and the REM stage are easily confused in classification. This may be due to some common features in the two stages. The classification of the N1 stage has always been a difficulty in the automatic sleep stage classification task, and the evaluation indicators are relatively low. On the one hand, the number of samples available for training in the N1 stage is small. Another factor is that the N1 stage is a transitional stage. It has multiple features before and after the transition stage, which increases the difficulty of the classification.

Table 3 shows the experimental results based on the Pz-Oz channel data, mainly including the relevant confusion matrix data and the corresponding evaluation indicators values. It is not difficult to see that the experimental results based on the Pz-Oz channel data and the experimental results based on the Fpz-Cz channel data in Table 2 show similar rules. The difference is that the experimental results based on Pz-Oz channel data have declined in terms of classification performance. This may have been caused by the different coupling degree between the electrode positions corresponding to different channels and the sleep area of the brain, because the same phenomenon has also appeared in other related studies.

The proposed model is compared with related research in this paper, as shown in Table 4. 

U-time [20] draws on the u-net model in the field of image segmentation. The sleep classification task is treated as a classic segmentation task, and the dice loss function is used. It has achieved good performance in multiple datasets, but CNN cannot have good memory for long time sequence. IITNet [21] is an intra- and interepoch temporal context network, which extracts the features of subepochs. ResNet50 and Bi-LSTM are used to learn network features, but ResNet50 has a very large number of parameters. DeepSleepNet is a representative research that uses the CNN‒RNN framework, but it cannot extract detailed features in the sleep stage. TinySleepNet [22] is an improvement and upgrade to DeepSleepNet; a data enhancement scheme and simpler network framework are adopted. The results of this method in Table 4 are reproduced based on the original paper code on the half-hour W stage data before and after sleep dataset. Tsinalis et al. [23] performed feature extraction, which requires manual feature design. Zhu et al. [24] used the attention mechanism to improve the CNN network. Tsinalis et al. [4] and Sun et al. [25] adopted the basic strategy of convolutional pooling for feature extraction without considering the relationship between states. Compared with other studies, CCRRSleepNet has achieved the highest performance or sub-high performance on multiple indicators. The results on the Fpz-Cz channel and u-time are basically same in the N1 stage, but the REM stage is the second highest. This is because u-time has carried out the data processing strictly, including quality control and other methods. Compared with [25] on the Pz-Oz channel, the performances in the N2 and REM stages are the second highest. However, the model in this paper increased by 16.64% in the N1 stage, which shows that the model in this paper is more robust. In addition, the authors of [25] did not perform experiments on the complete dataset, so there is a certain experimental deviation. In general, the method proposed in this paper has achieved better performance on both channels.

### 4.2. Ablation Experiment

In order to further verify the impact of relational inductive bias on sleep stage classification, ablation experiments are carried out to determine whether the model includes epoch-level CNN and epoch-level RNN. For the impact of frame-level CNN and sequence-level RNN on the network, many studies have shown their effectiveness, so no verification was made in this paper. In cases where the other experimental settings are consistent, the experimental results are as shown in Table 5. The table contains the corresponding performance of the network and its improvement compared to the baseline.

In Table 5, C-R means Frame-level CNN and Sequence-level RNN only; C-C-R means Frame-level CNN, Epoch-level CNN and Sequence-level RNN; C-R-R means Frame-level CNN, Epoch-level RNN, and Sequence-level RNN; and C-C/R-R means CCRRSleepNet. The network performance without epoch-level CNN and epoch-level RNN is used as a baseline for comparison in this paper. When the epoch-level CNN or RNN is added, the network performance is improved. This indicates that the features extracted by the epoch-level CNN and the epoch-level RNN are beneficial to the classification results. When epoch-level CNN and epoch-level RNN are added at the same time, the value of promotion is higher than when adding only one of them. This means that they have extracted different features. However, the value of the boost is not as large as the sum of the two individual boost values. This indicates that these two have some common features. These common features may consist of low-level features of the signal and noise. In addition, the epoch-level RNN has a higher increased value than the epoch-level CNN, because the frame-level CNN and the epoch-level CNN use the same type of relational inductive biases. To learn more abundant features for downstream tasks, matching relational inductive biases must be applied to the signal. This shows that “matching” means not only that different relational inductive biases are required to match at the same level of the signal, but also that the same kind of relational inductive biases must be matched with their own hyperparameters at different levels of the signal.

In order to explore the impact of nonrelational inductive biases on sleep stage classification, a comparative experiment on focal loss function and sequence-based sampling method was carried out. Since the training process of this model is unstable when the Relu activation function is used, no relevant comparison was made. The experimental results are shown in Table 6.

In Table 6, when focal loss is not checked, the cross-entropy loss function is used. Class balance refers to random upsampling of classes in the pretraining stage, and state transformation relationship balance refers to sequence-based sampling of data in the fine-tuning stage.

Through experimental comparison, we found that the network performance is more significantly improved by focal loss than by cross entropy loss. When the sequence-based sampling method is added to balance out the sleep state conversion relationship, performance is also improved. When class balance is not performed, the performance declines significantly if using the cross-entropy loss. This indicates that the nonrelational inductive biases of mismatch cause the neural network to fail to enter the optimal convergence point. We also found that, when all of the above strategies are used, the best results are not achieved. This may be due to the overfitting caused by the upsampling operation. In addition, the class balance will increase the training cost due to the use of upsampling.

### 4.3. Relational Inductive Biases Analysis

Different relational inductive biases are used in existing frameworks to extract the features of physiological signals, but they do not use matching relational inductive biases at all levels of the signal. This will lead to incomplete information extraction. The CCRRSleepNet proposed in this paper adopts matching relational inductive biases for all levels of physiological signals, and the extracted features are more complete. Table 7 shows the relational inductive biases applied in these levels and the corresponding features extracted.

Frame-level CNN uses a smaller convolution kernel to extract the basic properties of the signal, such as amplitude, phase, and slope. Epoch-level CNN uses a larger convolution kernel to extract waveform features in the signal, such as K-complex, spindle, and various rhythms. The epoch-level RNN learns the time-varying features among the basic attributes in different time periods of the frame level. Finally, the basic attributes from the frame-level CNN, the waveform features from the epoch-level CNN, and the time-varying features from epoch-level RNN are aggregated and used as the sequence-level input, and the sequence-level RNN is used to learn the transformation relationship of the sleep state. Deep learning uses relational induction bias for effective learning, but it is difficult for humans to fully and effectively explain the knowledge they have learned. For example, how the weight of deep learning is directly related to neuroscience theory. Nevertheless, the knowledge extracted by deep learning has this inevitable connection with the knowledge used by humans, but this connection is currently not completely clear. It is precisely because this method that humans cannot fully understand for the time being can discover potential knowledge that is temporarily unknown to humans, so it can replace humans to complete tasks that humans consider to be very complicated.

It should be pointed out that CCRRSleepNet does not need to strictly follow the construction method of this paper. This allows for the use of task-related relational inductive biases in different submodules and their extensions. Therefore, the CNN or CNN‒RNN model used in many other studies can also be considered a special simplified structure of CCRRSleepNet. At the same time, the CCRRSleepNet model provides a new way to construct a network structure for sequential signal processing.

## 5. Discussions

This paper designs a hybrid relational induction bias network CCRRSleepNet for sleep stage classification. The results show that the model can be applied to different EEG channels (Fpz-Cz and Pz-Oz) without changing the model structure and training algorithm. On the Sleep-EDF dataset, compared with the most advanced deep learning methods, it has achieved higher performance. It is verified by ablation experiments that the increased relational induction biases and non-relational induction biases can improve performance to varying degrees. Although the method proposed in this paper improves certain performance, our model still suffers from some limitations. First of all, the model only applies to the specific channels they have been trained on. We have tried using different channels of signals for training and testing, but the classification accuracy has dropped by about 10%. Although this performance gap can be reduced by fine-tuning, it is still slightly lower than the performance of training directly from scratch. Secondly, the model has been tested on a relatively small data set at present, and further research is needed. The Sleep-EDF dataset is a benchmark dataset for sleep stage classification, but it cannot represent all human sleep states. Especially the correlation between sleep and age is very high, and the data collected in this dataset only includes adults aged 25–34. Therefore, whether our model applies to other age groups still needs further research. Finally, like most papers focusing on deep learning technology, this paper is difficult to fully and effectively explain the knowledge it has learned. Nevertheless, the knowledge extracted by deep learning objectively reflects the inherent characteristics of neuroscience. It can provide further analysis support for scholars who focus on neuroscience research.

In practical applications, the proposed model can classify sleep stages only by collecting EEG signals at the same location. There are differences between real-world data and datasets, which can be fine-tuned to improve the suitability of the dataset or pre-training with larger datasets. In addition, this paper used 20-fold cross-validation to make a fair comparison with previous studies. In practice, three ways can be used to infer: (1) A fold weight file is directly used, because the training data and test data belong to the same distribution dataset in theory; (2) To improve the robustness of the model, all the data of the dataset can be used to train as a weight file for actual testing; (3) Multiple weight files can be used for integration. This paper will provide the open-source code of the model and a folded weight file on both channels for practical application.

## 6. Conclusions

A deep learning framework CCRRSleepNet for sleep stage classification is designed in this paper by introducing some inductive biases that match the task. By adding a frame-level CNN and an epoch-level RNN, more detailed relational inductive biases that match the task are introduced, which enhances the characterization ability of the network and effectively alleviates the performance limitation problem caused by the incompleteness of the feature extraction method. To further expand the feature extraction range and nonlinear ability of CNN, the MSACB is proposed, which can effectively improve the feature extraction ability of the network without significantly increasing the network parameters. At the same time, nonrelational inductive biases strategies are adopted in this paper such as data enhancement, data balance, activation function, optimization method, learning rate strategy, and loss function, which are more in line with sleep signals. The model proposed in this paper is tested on the Sleep-EDF dataset. In the Fpz-Cz channel without any preprocessing, an overall accuracy rate of 84.29%, an MF1 score of 79.81, and a Cohen’s Kappa coefficient of 0.78 have been achieved. In the Pz-Oz channel without any preprocessing, an overall accuracy rate of 84.29%, an MF1 score of 74.59, and a Cohen’s Kappa coefficient of 0.73 have been achieved. The method proposed in this paper surpasses almost all existing advanced classification methods. At the same time, ablation experiments are conducted on CCRRSleepNet to verify the effect of the matching degree of relational inductive biases and non-relational inductive biases on the model performance. In summary, the CCRRSleepNet model in this paper has superior classification performance in sleep stage classification tasks, and the model construction method proposed in this paper also provides a new option for solving timing signal problems.

## Figures and Tables

**Figure 1 brainsci-11-00456-f001:**
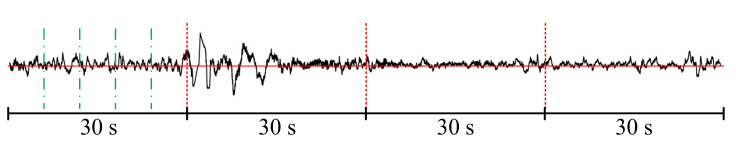
Time domain level division of physiological signals.

**Figure 2 brainsci-11-00456-f002:**
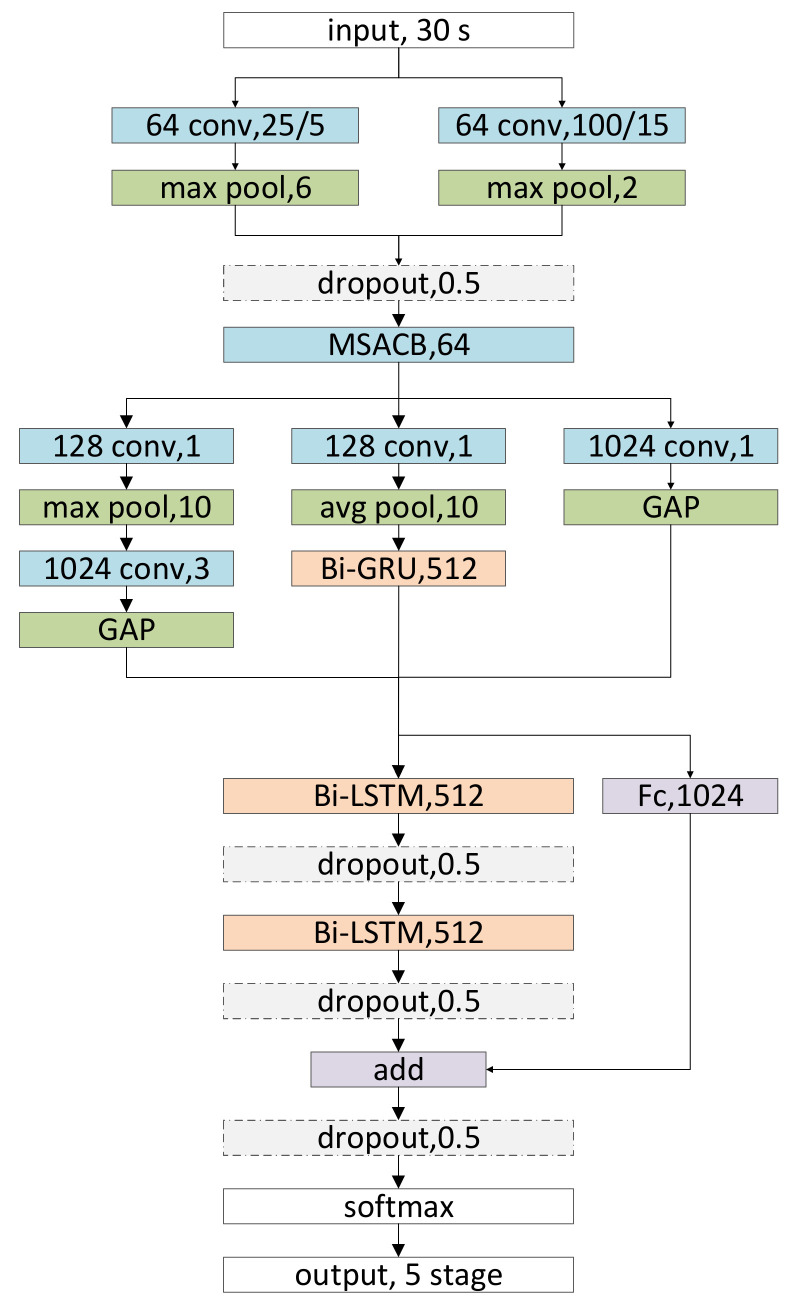
Overall architecture of CCRRSleepNet network. Conv is convolution, MSACB is multiscale atrous convolution block, GAP is global average pooling, Bi-GRU is two-way GRU network, Bi-LSTM is two-way LSTM network, Fc is full connection, and add means point-by-point addition.

**Figure 3 brainsci-11-00456-f003:**
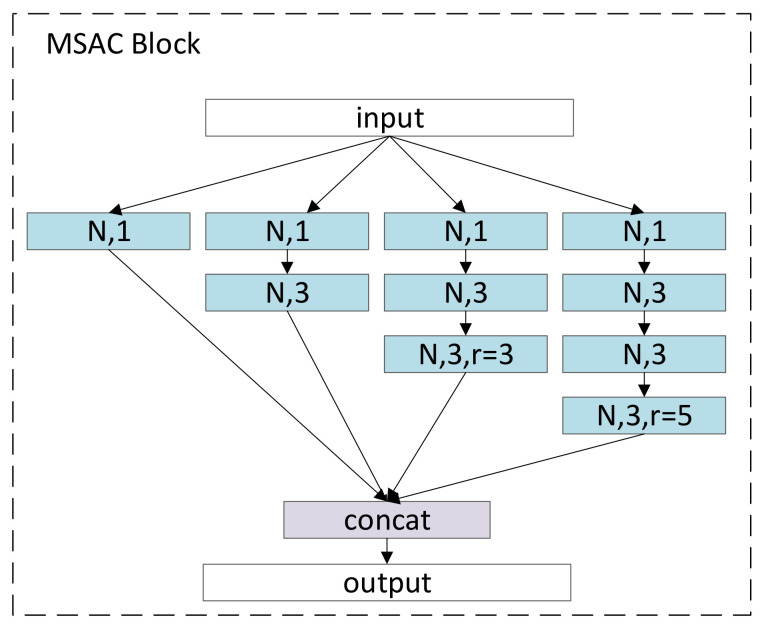
Multiscale atrous convolution block. Where r represents the dilated rate, and N represents the number of channels.

**Figure 4 brainsci-11-00456-f004:**
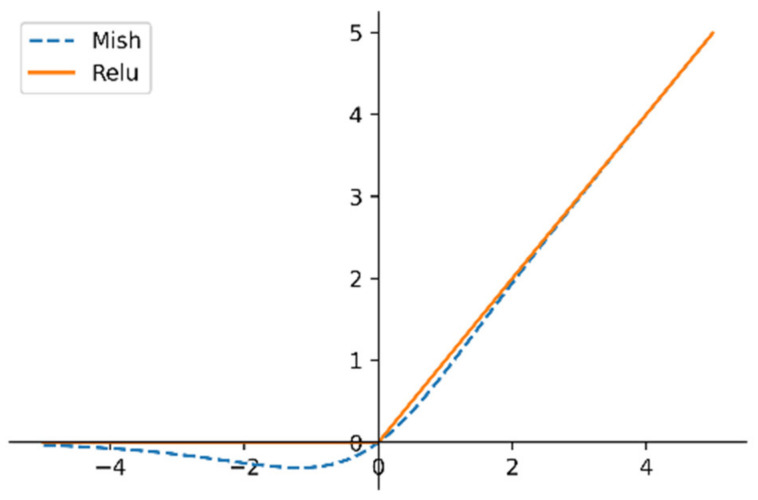
Comparison of Mish and Relu activation functions.

**Table 1 brainsci-11-00456-t001:** Sample distribution of Sleep-EDF dataset.

Dataset	W	N1	N2	N3	REM	Total
Sleep-EDF	7927	2804	17,799	5703	7717	41,950

W means wakefulness, REM means rapid eye movement, N1 means non-REM 1, N2 means non-REM 2, N3 means non-REM 3.

**Table 2 brainsci-11-00456-t002:** Performance of CCRRSleepNet on Fpz-Cz channel.

	Predicted					Per-Class Performance (%)		
	W	N1	N2	N3	REM	Pre	Re	F1
W	**6761**	781	183	21	181	93.06	85.29	89.01
N1	250	**1582**	579	4	389	47.77	56.42	51.73
N2	184	635	**15,638**	507	835	86.65	87.86	87.25
N3	19	8	747	**4919**	10	90.24	86.25	88.20
REM	51	306	900	0	**6460**	82.03	83.71	82.86

Bold represents the number that is correctly classified.

**Table 3 brainsci-11-00456-t003:** Performance of CCRRSleepNet on Pz-Oz channel.

	Predicted					Per-Class Performance (%)		
	W	N1	N2	N3	REM	Pre	Re	F1
W	**6728**	824	105	13	257	87.18	84.87	86.01
N1	495	**1313**	532	17	447	37.33	46.83	41.54
N2	179	821	**14,915**	1066	818	85.97	83.80	84.87
N3	19	22	1028	**4631**	3	80.74	81.20	80.97
REM	296	537	770	9	**6105**	80.01	79.11	79.56

Bold represents the number that is correctly classified.

**Table 4 brainsci-11-00456-t004:** Comparison of CCRRSleepNet with other methods.

Methods	Architecture	Test Epochs	EEG Channel	Overall Performance			Per-Class Performance (F1)				
				ACC	MF1	κ	W	N1	N2	N3	REM
U-Time [20]	C	41,950	Fpz-Cz	-	79	-	87	52	86	84	**84**
IITNet [21]	C-R	42,308	Fpz-Cz	84.0	77.7	**0.78**	87.9	44.7	**88.0**	85.7	82.1
DeepSleepNet [13]	C-R	41,950	Fpz-Cz	82.0	76.9	0.76	84.7	46.6	85.9	84.8	82.4
TinySleepNet * [22]	C-R	41,950	Fpz-Cz	83.6	78.7	0.77	86.8	49.9	87.4	86.4	80.6
Tsinalis et al. [23]	-	37,022	Fpz-Cz	78.9	73.7	-	71.6	47.0	84.6	84.0	81.4
Tsinalis et al. [4]	C	37,022	Fpz-Cz	74.8	69.8	-	65.4	43.7	80.6	84.9	74.5
Zhu et al. [24]	C	42,269	Fpz-Cz	82.8	77.8	-	**90.3**	47.1	86.0	82.1	83.2
CCRRSleepNet	C-C/R-R	41,950	Fpz-Cz	**84.29**	**79.81**	**0.78**	89.01	**51.73**	87.25	88.20	82.86
DeepSleepNet [13]	C-R	41,950	Pz-Oz	79.8	73.1	0.72	**88.1**	37	82.7	77.3	80.3
Sun et al. [25]	C	18,815	Pz-Oz	**81.0**	73.6	-	85.6	24.9	**88.9**	79.2	**86.3**
CCRRSleepNet	C-C/R-R	41,950	Pz-Oz	80.31	**74.59**	**0.73**	86.01	**41.54**	84.87	**80.97**	79.56

Bold denotes the best performance. C represents the CNN framework, C-R represents the frame or epoch level CNN sequence level RNN framework, and C-C/R-R represents CCRRSleepNet. The numbers in bold indicate the best performance indicators of all methods on the corresponding channel; * indicates reproducible performance.

**Table 5 brainsci-11-00456-t005:** The impact of relational inductive biases on network performance.

Relational Inductive Biases	Epoch-Level CNN	Epoch-Level RNN	ACC/Up	MF1/Up	κ/Up
C-R			81.78/baseline	75.82/baseline	0.75/baseline
C-C-R	√		83.16/1.38	77.14/1.32	0.76/0.01
C-R-R		√	83.59/1.81	79.02/3.20	0.77/0.02
C-C/R-R	√	√	**84.29/2.51**	**79.81/3.99**	**0.78/0.03**

Bold denotes the best performance.

**Table 6 brainsci-11-00456-t006:** The impact of nonrelational inductive biases on network performance.

Focal Loss	Class Balance	State Transformation Relationship Balance	MF1
			74.34
	√		78.75
√	√		79.21
√			79.46
√		√	**79.81**
√	√	√	79.66

Bold denotes the best performance.

**Table 7 brainsci-11-00456-t007:** Relational inductive biases of CCRRSleepNet model.

Level	Submodule	Relational Inductive Biases	Extract Features
Frame	CNN	Space invariance	Basic attributes
Epoch	CNN	Space invariance	Time-varying features
Epoch	RNN	Time invariance	Time-invariant features
Sequence	RNN	Time invariance	State transition relationship

## Data Availability

Data are available upon request.

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
