# Peer review of "CCRRSleepNet: A Hybrid Relational Inductive Biases Network for Automatic Sleep Stage Classification on Raw Single-Channel EEG"

_brainsci, 2021, doi:10.3390/brainsci11040456_

Round 1

Reviewer 1 Report

This is a well-written paper. Please find my comments below:

Please report some of the findings, especially Cohen’s Kappa agreement in summary.

Line #53-64, #73-78: Please report the performance of the models you reviewed here.

Please explain the abbreviations used in figure 2 in the caption of figure

How many epochs are used in sequence level? and why that number?

Could you please report the results of the model trained on the Fpz-Cz channel but tested on Pz-Oz? The reason I ask this is to know if the model is applicable to other EEG channels. Especially since the Pz-Oz channel is not the recommended channel of AASM.

The SleepEDF dataset is relatively small in size. Can Authors test their model on different datasets? Especially with different EEG recording frequency.

Line #352-354: Compared to which methods? Did they train their model on the same sample size?

If one wants to apply this model (CCRRSleepNet) on a different dataset or into a software to score the sleep stages of a sleep study, what are the weights that should be used? In this paper, the authors used k-fold approach, which means that the weights are different in each of the 20 runs. But what would be the final model? We shouldn’t forget that the final goal is to make an automated sleep scoring model (such as CCRRSleepNet), to replace with manual scoring. But right now, CCRRSleepNet needs to be trained on each new dataset, which means that we still need manual scoring to train it.

Please add a statement regarding the code availability.

Reviewer 2 Report

The draft demands substantial revision in every part and an extensive editing of English language.

The authors reported a deep learning EEG study for automatic sleep stage detection.

The study demands substantial revision and rewriting.

Below, I summarize my comments:

1) First of all, there are numerous publications combining DL algorithms with EEG recordings to solve this open problem in sleep. 

The authors should review these studies and describing why the proposed architecture differs from those.

2) The authors should justify why they design DL architecture in such a way.

     They should give also an example of an input and what kind of information is learned from an EEG epoch.

3) The authors didn't report details regarding EEG preprocessing steps.

    Did they use the dataset without artifact correction, in broadband frequency bands, within which frequency range?

4) The range of age is broad and for that reason, there is definitely an effect   of age by mixing epochs per sleep stage across the cohort.

   It is preferable to somehow deal with this issue by dividing the group into two age cohorts or working per subject.

5) You didn't report the features learned from the DL scheme and how these features are related to our knowledge regarding sleep process.

6) Why did you use the 2 sensors reported in the methods part?

7) You must describe every methodological part in every detail, justifying your selection, reporting the EEG features and give a neuroscientific interpretation of the findings.

Round 2

Reviewer 1 Report

Regarding point 4 of response to reviewer 1 comments: Please explain how did you use the 25 epochs window, i.e., 12 epochs prior and 12 epochs after? or 24 epochs prior?...   Regarding point 5 of response to reviewer 1 comments: Please add this explanation to the limitation section of the article. I.e., the model is only applicable on the specific channel that they have been trained on.   Regarding point 6 of response to reviewer 1 comments: Please add this point to the limitation section of the article. I.e., the model has only been tested on a relatively small size dataset and further investigations are needed.   Regarding point 8 of response to reviewer 1 comments: Please add these explanations to the paper and specify which weight file would be available publicly.      

Reviewer 2 Report

The authors addressed the majority of my comments.

However, I have to insist on major drawbacks.

1) First of all, the authors didn't correct the signals from artifacts.

    This is a drawback and limitation of the study.

    Your approach analyzed directly the amplitude of EEG time series.

     For that reason, there is an effect of artifacts on the final performance.

2) The range of age is broad. So, your approach must be work also on subgroups of the cohort. The effect of age is not only between children and adults but also during the lifespan.

A.If you train your DL scheme with a hypnogram from the younger individuals

and test it with the older subgroup, you can justify my concern.

or you can use a secondary dataset for external evaluation. 

B.You reported: 'First, the Sleep EDF dataset was divided into k subsets, and data from the same individual were not included in multiple subsets. The kth subset was used for testing, and the rest were used as training data.'

Which strategy did you adopt to not include epochs from the same individual in multiple folds?

You have to explain it explicitly.

3) There is a nice review where you can compare your approach with others using DL.

Automated Detection of Sleep Stages Using Deep Learning Techniques: A Systematic Review of the Last Decade (2010–2020)

4) A Discussion part is missing from the text.
